# Hardware-in-the-Loop Test Bench for Simulation of Catenary–Pantograph Interaction (CPI) with Linear Camera Measurement

**DOI:** 10.3390/s23041773

**Published:** 2023-02-04

**Authors:** Antonio Correcher, Carlos Ricolfe-Viala, Manuel Tur, Santiago Gregori, Mario Salvador-Muñoz, F. Javier Fuenmayor, Jaime Gil, Ana M. Pedrosa

**Affiliations:** 1Instituto de Automática e Informática Industrial, Universitat Politècnica de València, Camino de Vera s/n, 46022 Valencia, Spain; 2Instituto de Ingeniería Mecánica y Biomecánica, Universitat Politècnica de València, Camino de Vera s/n, 46022 Valencia, Spain

**Keywords:** catenary–pantograph, computer vision, linear camera, hardware-in-the-loop

## Abstract

Catenary–pantograph contact force is generally used for assessment of the current collection quality. A good current collection quality not only increases catenary lifetime but also keeps a stable electric supply and helps to avoid accidents. Low contact forces lead to electric arcs that degrade the catenary, and high contact forces generate excessive wear on the sliding surfaces. Railway track operators require track tests to ensure that catenary–pantograph force remains between safe values. However, a direct measure of the contact force requires an instrumented pantograph which is generally costly and complicated. This paper presents a test bench that allows testing virtual catenaries over real pantographs. Therefore, the contact point force behavior can be tested before the track test to guarantee that the test is passed. Moreover, due to its flexibility, the system can be used for model identification and validation, catenary testing, or contact loss simulation. The test bench also explores using computer vision as an additional sensor for each application. Results show that the system has high precision and flexibility in the available tests.

## 1. Introduction

According to the Eurostat report: “Key Figures on Europe: 2021 Edition”, transport has been the primary energy consumption sector in the European Union (EU) since 1996. In 2019, transport consumed over 130 tons of oil equivalent. That represents 30.9% percent of the overall energy consumed in the EU [1]. The European Green Deal calls for a 90% reduction in transport emissions by 2050, which necessarily involves an impulse of electric railway transport.

Electric railway trains and trams need a continuous electric flow, usually provided by the overhead line equipment commonly known as a catenary. A conductor mechanical device (a pantograph) is often attached to the locomotive’s roof to collect electricity from the catenary. The catenary–pantograph interaction (CPI) force is essential for appropriate current collection and catenary maintenance. If the contact force is too high, excessive friction will cause wear on the sliding components (catenary contact wire and pantograph collector strips). On the other side, if the contact force is too low, the contact can be occasionally lost, which generates poor electrical collection and arcs (also leading to wire degradation). Therefore, each electric locomotive must meet requirements according to standard UNE-EN 50317 [2] to get permission to circulate on an EU railroad.

In view of the CPI significance, much research has been done in recent years. Advanced mathematical models [3,4,5,6,7,8,9,10] have realistically allowed the study of CPI. These models have been used for simulation and exploring several problems affecting CPI, such as wire irregularities, track deficiencies, catenary mounting errors, or aerodynamic loads caused by the wind or by the incoming flow due to train speed. Contact irregularities are explored in [5]. The authors investigate the effect of contact wire irregularities on the CPI force and conclude that these irregularities directly impact the CPI. The influence of vehicle body vibrations due to track irregularities on CPI is described in [11,12]. Installation errors on the catenary and their effect on CPI are presented in [10,13]. An assessment of the catenary’s wind deflection is shown in [14]. The effects of aerodynamic forces over CPI are intensely studied in [15].

Trains must pass costly on-track tests to gain a registration certificate, so it is important to ensure that the current design fulfills the requirements. Recent advances in catenary models and mechatronics have allowed the development of test benches for realistic off-track pantograph testing. The approach used to design those test benches is the so-called hardware-in-the-loop (HIL) simulation. In the HIL approach, an experiment over a system is performed by a hybrid simulation where one part of the system is real, and the other is simulated. HIL is a technique used in many fields, such as automotive, energy, or electrical systems. There are a few HIL test benches reported all over the world to test the CPI. Test benches differ in the type of catenary models they can run, the technology used to simulate the CPI, and the kind of simulations they can perform: open-loop or closed-loop. The open-loop approach generates a force pattern on the pantograph to test the response. On the contrary, the closed-loop approach measures the force on the contact point to provide feedback to a catenary model that generates the movement at each sample time.

Regarding open-loop proposals, this kind of testing allows verifying the wearing and basic performance of the pantographs, but they do not test the real response of the pantograph against changing conditions. The test bench developed by MTS [16] (Eden Prairie, MN, USA) can reproduce a catenary load over a pantograph in open-loop mode. Therefore, it cannot perform HIL simulations but reproduces movements over the top and bottom of the pantograph. The actuator of the top of the pantograph constantly turns, thus simulating the train movement. This system performs vertical movements up to 10 Hz and can transmit power through the contact point (up to 2 kA). 

Another example is the test developed at the Railway Technical Research Institute [17] (Hino-shi, Tokyo, Japan). The working principles are very similar to the test bench by MTS, but this device can achieve higher speeds in rotation and up to 1 kA of current. These authors are currently developing a dynamically substructured test bench to perform closed-loop CPI testing [18,19]. This test bench uses an outer-loop substructured controller to minimize the difference between the predicted and measured contact points. Preliminary results point to good performance on error reduction, but the hydraulic actuator limits the speed of the tests.

In the case of closed-loop test benches, the first proposals [20,21] used a truncated modal approach to simulate the virtual catenary and two servo-hydraulic actuators to interact with the real pantograph. The lower actuator was used to generate the locomotive motion on the bottom of the pantograph. According to the virtual catenary simulation, the upper actuator moves the pantograph collector head. The response time of the actuators was 20 ms (up to 50 Hz). Another HIL simulator and its improvements are presented in [22,23]. This test bench generates the movement of the CPI on the contact strips of the pantograph using an electrohydraulic actuator. This actuator guarantees a bandwidth of 25 Hz. The simulation of the catenary runs on a real-time platform and includes non-linear dropper behavior. More recently, [24] presented a more advanced catenary model with 3D Euler-Bernoulli beams. The catenary gives the position and velocity references to a robot arm, generating the CPI movement on the pantograph. This system includes a controller to guarantee HIL simulation stability at the cost of generating fake force to counteract the effect of high frequencies in the variables. Because the robot consists of electric actuators, this system can work faster than the others reaching frequencies up to kHz.

The test bench at the University of Huddersfield [25] is the biggest of the reported closed-loop HIL test benches. It includes the capability of open and closed-loop HIL simulations, simulation rates up to 100 Hz, lateral movement to simulate catenary stagger, and bottom inclination of the pantograph (up to 5°).

The Universitat Politècnica de València in Spain has developed a test bench [26,27] capable of real-time simulating finite element catenaries with sample times up to 2 ms. A detailed description of this test bench and its improvements is shown in Section 2.

Another trend in CPI assessment that has been developed recently is the use of computer vision [28,29,30,31,32,33,34,35,36]. Cameras give contactless information about CPI that can be used for many purposes. Examples of applications are identifying catenary characteristics [30] or anomalies [31]. The main application of computer vision techniques is non-contact detection in CPI [29]. In this field, electric arc detection is a research hotspot [32] due to its direct application to condition monitoring [33]. Primary efforts in CPI computer vision assessments are oriented to deal with complex backgrounds [35], which complicates the labeling and processing of the images. Works in this direction take advantage of the modern high-capacity processors that run complex deep-learning tools [29,36].

Besides CPI monitoring, computer vision is gaining prominence as a condition-monitoring basic tool in the railway industry. Many commercial solutions analyze the pantograph when the train reaches the position of the camera. Examples of these solutions are Pantobot 3D [37] by Camlin Rail (Lisburn, N. Ireland) or Pantoinspect [38] (Copenhagen, Denmark).

This work presents a test bench to perform HIL simulations of virtual catenaries with real pantographs and explores the synchronization of the CPI simulation with computer vision. The HIL test bench simulates the CPI over a train at different speeds and can make real-time measures of the contact force. A synchronized linear camera will take images of the CPI that can be analyzed offline. This research will prove the feasibility of using contactless sensors in pantographs for performing the on-track homologation train tests. It is important to note that this work does not pursue developing a computer vision application directly applicable to real trains but to prove the technique’s feasibility in a laboratory environment.

After this introduction, Section 2 is devoted to describing all the components that form the test bench and the processes required to perform a HIL test. The results are presented in Section 3, which includes a validation of the HIL setup, an example of a HIL test, the calibration of the linear camera, an experiment with contact loss, and some words about pantograph identification. Finally, discussion and future work are addressed in Section 4.

## 2. Materials and Methods

The HIL CPI simulation takes place on the system shown in Figure 1.

The system consists of several elements capable of precisely simulating the movement of the contact point between a catenary and a pantograph. A scheme of the main components and their interactions is shown in Figure 2.

A linear motor simulates the catenary position. As it moves, it pushes the pantograph, which reactively pushes the motor slider. Two force sensors measure the reactivity to the motor (acting as a catenary) and send the value to a mathematical model of the catenary running in a virtual environment. The model uses the force as an input and computes the next state of each point in the catenary. Considering the train’s speed, the model selects the position of the following CPI point and sends it to the motor controller. Then the motor moves to this position and closes the simulation loop. Additionally, a linear camera takes images of the CPI point synchronized with the force measurements. The aim of the linear camera is to measure the vertical displacements of the pantograph head and the contact wire.

The main elements of the test bench are the linear motor, the real-time controller, the virtual catenary, the sensors, the pantograph, and the linear camera.

The linear motor

A motor drive (LINE1450) from LINMOT (Spreitenbach, Switzerland) controls a linear motor (LINPS10-70x240U-BL-QJ), allowing a maximum velocity of 5.4 m/s and a maximum force of 1650 N. It implements a closed-loop PID control strategy over the position of the slider. The position reference for the motor can be changed through EtherCAT communications. The control command set for the drive includes several reference generation modes. The selected working mode is position and velocity stream which is thought to be used in high-precision machines. Therefore, the system avoids generating parasite frequencies and discrete steps leading to instability.

2.The controller

A real-time device “CompactRIO 9040 (CRio) from National Instruments (Austin, TX, USA), controls the flow of the application. It includes a Dual Core 1.30 GHz processor, 2 GB RAM, Ethernet and EtherCAT communications, and several IO modules. The tasks assigned to the controller are: to guarantee the real-time flow of the application, receive the computed contact point positions from the virtual catenary, generate the control commands for the driver, measure and communicate the force on the pantograph to the virtual catenary, and save the experiment results. This device includes a real-time internal clock signal to generate real-time looped applications.

The controller can be set into four program modes. The Real-Time Mode allows the easy development of real-time applications with control loops up to 5 kHz. The Real-Time Scan Mode allows synchronous input-output updates with rates up to 1 kHz. It is thought of as real-time application development, including multiple synchronized loops. The FPGA Mode is thought to develop high-performance applications in the range of MHz, such as analog streaming, high-speed control loops, or high-rate signal acquisition and processing. The Hybrid Mode allows the use of the Real-Time Scan Mode for the main application in the CRio core and the FPGA Mode in some modules. This programming mode helps acquire and filter signals with high-speed requirements (such as acceleration) and integrate the measures with a complex real-time application.

3.Virtual catenary

It is essential to have some flexibility over the code to test different settings of the same catenary in different experiments. Moreover, it is also mandatory to consider a catenary model as realistic and efficient as possible to be solved in real-time. To fulfill these requirements, we use the analytic catenary model proposed in [8] and a periodic catenary finite element model [27]. The code of the catenary model runs on an (Intel^®^ Core™ i9-9900K CPU, 3.6 GHz, 64 GB RAM) PC that allows high-speed processing. Although these models are very suitable for dealing with delays in the HIL experiment, they only account for the steady-state response of the system and need to be combined with the iterative strategy developed in [26] to achieve convergence in a HIL test. However, the current work is focused on using a complete catenary finite element model, solved by the real-time numerical strategy proposed in [39], that will consider more realistic features such as overlaps or span length and dropper arrangement variability in a catenary section.

4.Sensors

Besides the camera, the system collects position, force in the contact point, and acceleration. The motor position sensor is built into the motor casing and has a precision of 0.005 mm. The force is measured with two force transducers (HBM C9C, Darmstadt, Germany) located in the two contact strips of the pantograph. Three accelerometers (Kyowa AS-5GA, Tokyo, Japan) can be located at different points of the pantograph and are measured via high-speed sampling modules in the CRio. When using the accelerometers, the CRio is set into the Hybrid Mode to allow FPGA filtering of the accelerations.

5.Pantograph

The pantograph is an Einholm DSA^®^ 380.03 from Stemman-Technik GmbH (Schüttorf, Germany). It has a single-arm and allows train speeds up to 380 km/h. A pneumatic system allows adjusting the static force on the CPI with the pressure of the compressed air intake. Table 1 shows its main characteristics.

6.Linear camera

The computer vision system includes a linear camera, a lens, and a frame grabber to speed up image acquisition. From the framing point of view, the approximate working area will be 100 mm separating the pantograph’s upper part from the lower part of the catenary, where the direct LED light sources have been installed. It is essential to consider the image capture speed in frames per second, which is decisive since it must satisfy the restrictions imposed by the system’s working frequency of 500 Hz.

Once the measurement characteristics have been defined, the camera and lens requirements can be determined. Assuming that it is necessary to detect movements of the order of 0.015 mm in a 100 mm frame, the resolution required for the camera can be established as follows:(1)$$\mathrm{Resolution}\geq \;\frac{100}{0.015}=6666.7\;\mathrm{pixels}$$

From the point of view of capture speed, the camera-to-computer connection is an important aspect. Each pixel is stored in one byte, and it is necessary to send at least 6667 pixels in 1 ms. Therefore, the connection speed between the camera and the processing equipment must be at least 6.7 Mb/s.

With all these parameters in mind, it is possible to decide which camera to use. From the point of view of the sensor format, we can use area or linear cameras. The area camera acquires two-dimensional image information during the capture time of a frame. The linear camera has a sensor with a single line of sensitive elements and requires movement relative to the scene to obtain a 2D image. The linear camera has a higher resolution and lower cost, which makes them suitable for measurement applications. Area cameras have a lower resolution per column than linear cameras.

The selected camera is an Aviiva from Teledynedalsa (Waterloo, ON, Canada). The camera has a resolution of 8192 pixels. Moreover, two direct light sources are installed in the pantograph and the motor (acting as the catenary), which will be used as markers to facilitate image processing.

The direct lights installed on the pantograph and in the motor are framed to project the light on the sensitive element’s column, as shown in Figure 3. A frame grabber allows Camera Link communication and ensures the system is deterministic regardless of the operating system used to guarantee the necessary frame rate. Moreover, a 50 mm F2.8 lens has been installed to illuminate the sensor correctly. With this lens, the camera can be placed at 200 mm of the setup, as shown in Figure 3, so it generates a minimum distortion in the image.

7.Interface

The control application includes a graphical interface allowing the complete tracing of the experiment. It includes signal visualization, signal spectrum analysis, span response, and data saving. Figure 4 shows the interface.

### 2.1. HIL Process

The CRio is a real-time device capable of generating multiple synchronized loops. Its internal clock has an accuracy of 200 ppm (parts per million). Therefore, this device is used as the core of the HIL simulation. The CRio launches three loops simultaneously (Figure 5). The first loop interacts with the virtual catenary, the second loop generates a trigger signal to synchronize the camera images, and the third loop communicates with the motor drive.

The first loop starts the experiment by measuring the current force. Then it sends the force to the VC through an ethernet UDP message. The PC computes the following contact point position and sends it back to the CRio. The third loop starts simultaneously and sends the current recorded CP as a position reference to the motor drive so that the motor moves to that height. As shown in Figure 5, the contact point sent to the motor corresponds with the computation of the VC model’s previous iteration. The motor movement generates a force over the pantograph, which is measured, thus repeating the loop.

The CRio implements UDP stable real-time communications with the VC with one millisecond or higher sample time. The driver communicates with EtherCAT in real-time, but it takes two milliseconds to update the internal variables. Because shorter sample times increase VC stability, the communication process is split into two synchronized tasks. Therefore, the CRio generates two synchronized real-time loops for each side of the experiment: a 2 ms loop for the VC and a 2 ms loop for the drive.

### 2.2. Virtual Catenary Computing

The VC side of the controller is executed each 2 ms deterministically. Although ideally, the system reacts to the force immediately, in the HIL simulation, some delays must be considered. The Virtual Catenary algorithm considers the system’s delays to compute the CP to be set as a reference for the motor drive (see for example [27] for details of the algorithm). The force signal is conditioned with an amplifier (NEC AS1201 AC Strain) that filters and increases the voltage levels. The hardware filter can be set to 10 Hz, 30 Hz, 100 Hz, 300 Hz, or no filtering to avoid high-frequency components in the force. This filtering process delays the force signal ranging from 20 ms to 0 ms, depending on the selected filter. This force measure gets into the VC algorithm, which computes the dynamic response of all the points in the catenary domain. Then, the catenary sends the appropriate CP position within the application period (2 ms) to guarantee delay compensation; see Figure 3. In the next application cycle, the CRio performs an EtherCAT writing in the motor drive to change the reference position of the motor. The motor moves to that position at that moment with a settling time of 3 ms. Therefore, the pantograph would meet the computed position three steps after measuring the force if the filter is set to 300 Hz or above.

### 2.3. Computer Vision System Configuration and Image Processing

The computer vision system consists of optics and a linear camera. Light goes through the lens and projects into the camera sensor for acquisition. The captured image allows for measuring the desired parameters of the pantograph experiment. The camera is chosen according to the characteristics needed to satisfy the measurements. In this case, the chosen camera described previously has 8192 pixels because the aim is to detect small movements in the order of tenths of millimeters. Since the camera framing is 100 mm, the range of small movements that it is able to detect is 0.012 mm.

Detecting the pantograph and wire in the image is done with markers to increase the efficiency of the computer vision-based measuring system. In this case, two direct light sources are installed, one in the pantograph and the other in the motor representing the catenary contact wire, as shown in Figure 3. Direct light saturates the pixel where it points; consequently, a grayscale value (0–255) is enough to measure the position of the lights in the frame. Using a grayscale image drastically reduces the data per frame and the image processing time.

With this setup, each experiment consists of a set of linear frames representing the position of the pantograph and the wire in a grayscale linear image at each HIL simulation step. Linear image frames are arranged in columns to build a 2D image that represents the experiment’s performance, as shown in Figure 6.

The computer vision algorithm that extracts information from the image is crucial to obtain the measurement of the pantograph and the wire in each HIL simulation step. Each linear frame is processed as follows:Pixels belonging to pantograph and wire are detected using image thresholding. The result is a boolean image.Selected pixels are grouped into blobs to analyze their features. Two blobs are detected that represent the pantograph and the wire.In this case, the centroids of both blobs are computed to determine their location in the frame.

### 2.4. Additional Experimental Capabilities

The test bench is thought to simulate virtual catenaries over a pantograph. Nevertheless, the application architecture offers the possibility of performing other experiments. The VC module can be changed to generate any signal, and the sensors will measure the response of the pantograph with the same application flow shown in Figure 5. Some experiment possibilities are pantograph identification, HIL simulation validation, data generation for artificial intelligence training, camera calibration, or contact loss simulation.

## 3. Results

This section shows some results of experiments performed with the system.

### 3.1. HIL Simulation Validation

Catenary models should be tested for stability before they can be used for HIL simulations. To verify the performance of each catenary, we test them offline at different speeds interacting with several pantograph models. If the VC has stable performance, it can be tested on the test bench. It is essential to verify if the simulated VC behavior will be reproduced on the test bench. Therefore, the VC can be used with other simple models besides the pantograph for validation purposes. A simple way of validating is placing a well-known system to interact with the VC and study the simulated and real curves. Figure 7 shows a single mass system attached to the motor instead of the pantograph.

For example, the periodic finite element model catenary presented in [27] was tested as explained, showing low error and, thus, validating the HIL simulation setup. Figure 8 shows a comparison of the contact force for a span between the simulation and the test.

Figure 9 shows the simulation of the same catenary with a lumped-mass pantograph model compared to the data obtained from the HIL simulation. Because of the validation of the HIL procedure shown in Figure 8, the pantograph model does not match perfectly with the real pantograph dynamic behavior. Pantograph modeling is a challenging task due to the non-linearity of several components and is a topic of ongoing research.

### 3.2. Virtual Catenary Simulation

Stable VC can be HIL simulated in the test bench. Figure 10 shows the HIL simulation of a periodic catenary model with the pantograph until the stabilization of the HIL iterative process.

The models are sensitive to high contact force changes at the start. Therefore, the amplitude of the catenary movement is only partially generated from the beginning. As shown in Figure 10a, the VC linearly increases the amplitude of the contact point position to guarantee a safe HIL simulation.

The real-time controller generates a trigger signal that synchronizes the contact point generation with the camera firing to obtain a set of grayscale values for each pixel and simulation time. Figure 11a shows the image of a HIL catenary simulation on its rising process. Figure 11b zooms into a section of an experiment where the HIL simulation steady state was established.

### 3.3. Camera Calibration

The camera is calibrated to obtain measurements in the image in millimeters. For this purpose, the system generates a 600 mm ramp to move the pantograph and to study this movement in the acquired images. The aim is to establish the ratio between pixels and millimeters in each frame. Figure 12a shows the generated pattern, and Figure 12b the acquired image. Each column represents one step of the calibration process, and the two white areas represent the position of the pantograph and the wire in the image. As mentioned before, each blob’s centroid defines the pantograph and the wire positions in the image. The red line in Figure 12b shows computed values in the image. Since we are measuring the distance between the pantograph and the wire, the calibration process computes the ratio pixel/mm. Moreover, the variation of this ratio over the length of the pixels line is also studied. Some optics distort the image, and this effect should be considered to obtain accurate measurements. In this case, the ratio has proved to be constant.

### 3.4. Contact Loss Simulation

Contact loss [28,40,41,42] is a well-known cause of catenary wear and can cause severe problems with the power supply and the safe running of trains. Because of the arcing phenomenon, cameras quickly detect the contact loss effect on the current transmission through the pantograph [43,44].

The test bench allows the HIL simulation of catenaries at different speeds and configurations, thus allowing the simulation of mounting errors, low pantograph pressure, and other effects that might lead to contact loss. Because the system is not electrified, the arcing phenomenon will not be produced. Therefore, the study of contact loss is restricted to any mechanical observation. The studies that can be performed will be oriented to the prognosis of contact loss by monitoring the movement of several parts of the pantograph. According to the standard EN 50367 [45], a contact loss will be considered if the electric arc is longer than 5 ms. Thus, a sampling rate of 2 ms is enough to capture this phenomenon. Figure 13a shows a part of an experiment where contact loss occurs. Contact loss can be detected by the force sensors when the interaction force reaches 0 N (Figure 13b). The complete experiment is shown in Figure 14, where the centroids of the two light beams detected by the camera are recorded to match the contact loss times.

### 3.5. Pantograph Identification

Section 3.1. showed that pantograph model identification is an interesting topic to be researched. The problem can be approached from different perspectives [46]. The test bench can generate any contact signal to study the response of the pantograph and fit a model, for example, chirp signals, including wide frequency content. Figure 15a shows a chirp signal, and Figure 15b the force response.

## 4. Discussion and Future Work

As a key sector within the transport business, the railroad industry is constantly modernizing its elements and systems. The rising requirements in passenger and freight transport lead to increments in train speed, which must be achieved together with the safe operation of both the train and facilities. As it has been proved to be a recent trend in the general industry, HIL simulation is being introduced in the railroad sector. HIL simulation allows the realistic testing of elements in a safe environment. Therefore, the development of systems and their commissioning is faster and safer.

In this context, this paper presents a test bench to perform HIL simulations of CPI that can be used for several lines of research, and it improves some aspects of facilities with the same purpose worldwide. The test bench allows the safe operation of the VC with sampling times up to 2 ms (500 Hz), which significantly improves the reported sampling times in other HIL test benches. Moreover, the electric actuator allows faster dynamics generation than hydraulic actuator devices [21,23].

One of the potential problems of HIL tests is the lack of stability in closed-loop. The interaction between VC models and real pantographs excites the pantograph in frequencies that can destabilize the catenary model [23]. The generation of high displacements produces high forces dangerous for the system’s safety. The frequencies that can destabilize the HIL simulation depend on the actuator technology, the VC model, and the sampling rate. Some of the reported test benches have solved the problem with external control. The goal is to stabilize the HIL process by adding an external actuation that modifies the CP forces. That action can be physical [23] or virtual [24]. From the authors’ point of view, this kind of control helps in the simulation stability but includes a force not present in the natural operation of the CPI. Because the main objective of a HIL simulation is to reproduce the CPI, adding forces that are not present in the real operation forces the VC to operate in a distinct way from what the real catenary will do. To validate the HIL simulation with external actuation, the actuation must trend to zero or remain in low values compared to measured force values.

Regarding flexibility in experimentation, the test bench at the University of Huddersfield [25] also has the capability of CPI HIL closed-loop simulation and pantograph identification. Moreover, it includes lateral movement to reproduce the catenary stagger and actuators at the bottom of the pantograph to simulate the railway configuration. Nevertheless, the actuators of the test bench are limited to the maximum frequency of the hydraulic system.

Therefore, the designed test bench allows the HIL closed-loop simulation of catenaries with a bandwidth of 500 Hz. It also allows catenary validation and pantograph identification. The test bench integrates computer vision as a sensor in the system, which allows the exploration of this kind of technology in HIL CPI testing. To improve the experimental capabilities, future work will address the implementation of some improvements. The lateral movement of the vertical actuator is important to reproduce the catenary stagger and study its effect over the pantograph dynamic. Therefore, a lateral actuator will be included. Another interesting research topic is the system behavior in curved paths. A bottom actuator will be placed to include car-body movement in the HIL process. The linear camera has proved to give interesting information. Therefore, a 2D camera will be placed to study the overall behavior of the pantograph. This new system will extend the CPI monitoring to other pantograph areas.

Finally, delays have also been detected as an important cause of instability. Thus, a more robust VC real-time simulator will be placed with data acquisition and generation capability to reduce as much as possible measure delays.

## Figures and Tables

**Figure 1 sensors-23-01773-f001:**
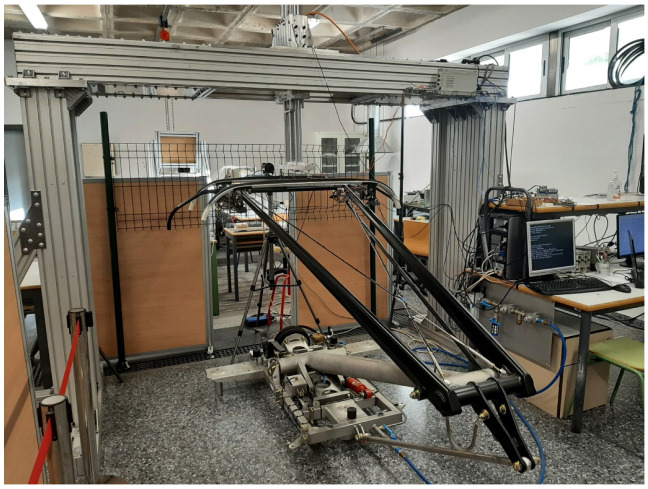
Test bench developed for the HIL CPI simulation.

**Figure 2 sensors-23-01773-f002:**
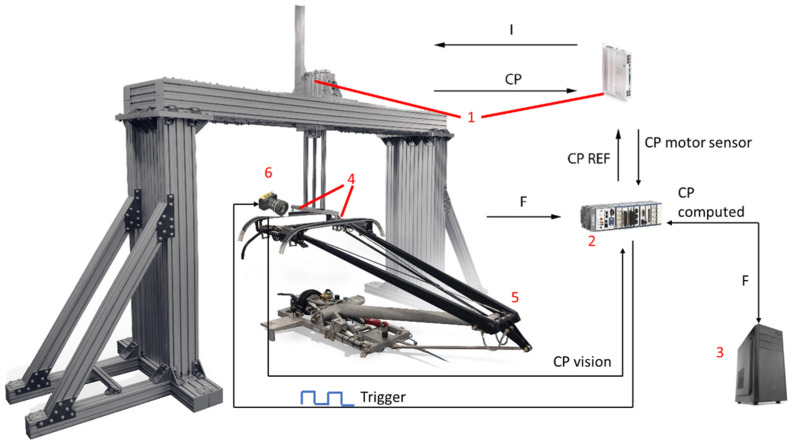
Scheme of the test bench components. 1. Linear motor and motor drive; 2. Real-time controller; 3. Virtual catenary; 4. Sensors; 5. Pantograph; 6. Linear camera.

**Figure 3 sensors-23-01773-f003:**
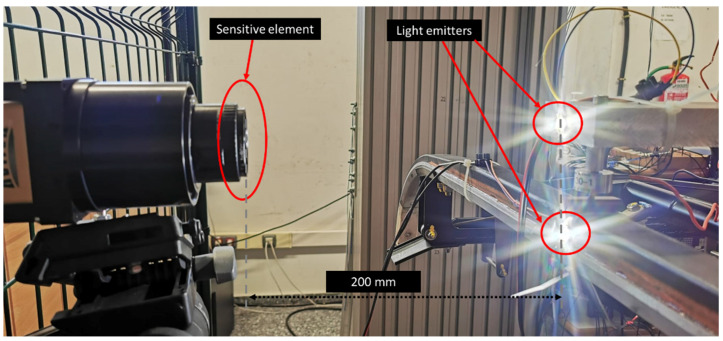
Linear camera setup.

**Figure 4 sensors-23-01773-f004:**
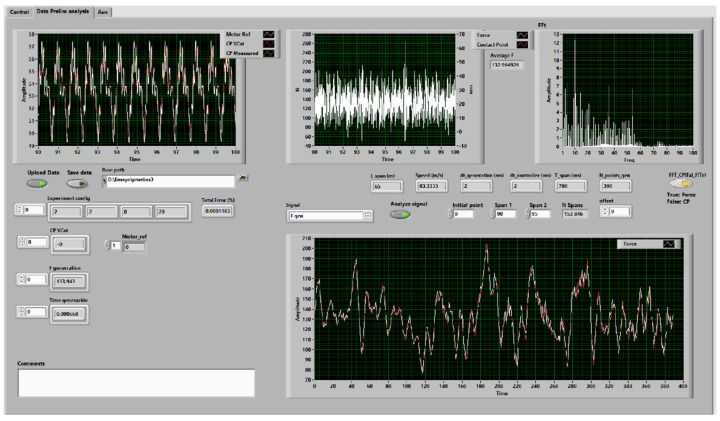
Application interface.

**Figure 5 sensors-23-01773-f005:**
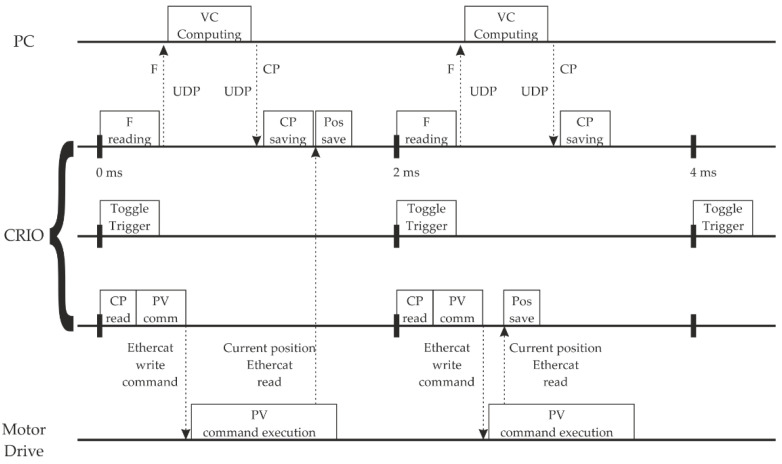
Software workflow.

**Figure 6 sensors-23-01773-f006:**
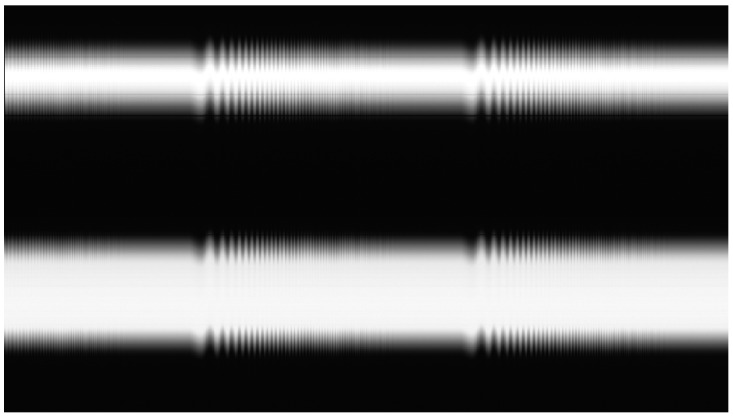
Linear image frames are arranged in columns to build a 2D image that represents the performance of the experiment.

**Figure 7 sensors-23-01773-f007:**
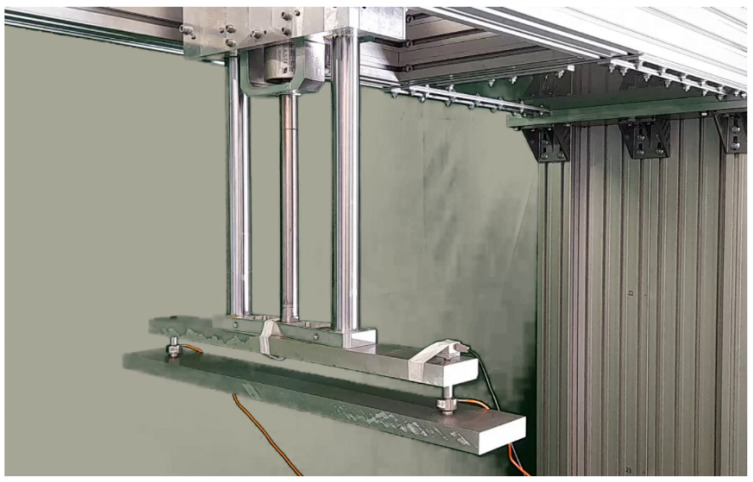
Single mass system.

**Figure 8 sensors-23-01773-f008:**
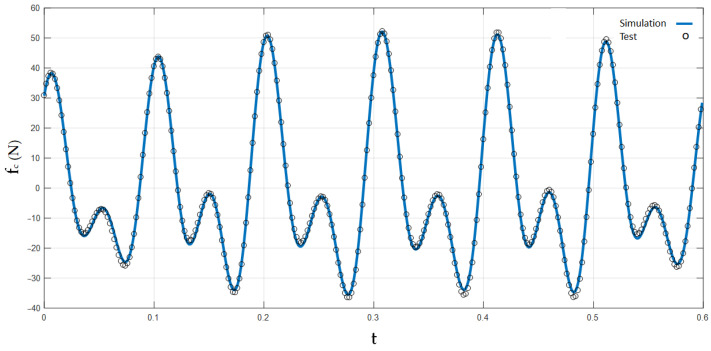
Validation of a periodic finite element catenary model.

**Figure 9 sensors-23-01773-f009:**
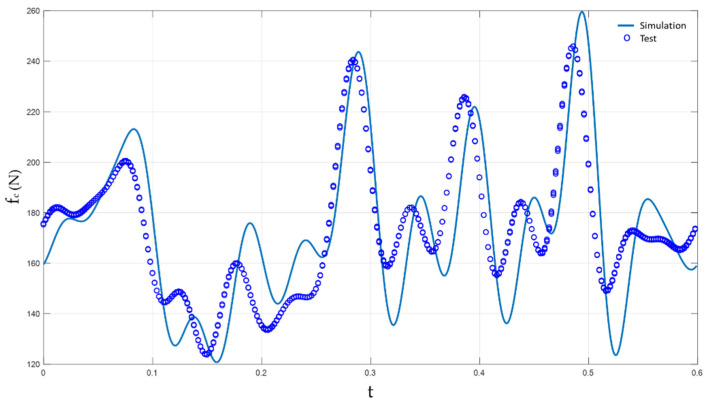
HIL simulation of a periodic catenary with a pantograph.

**Figure 10 sensors-23-01773-f010:**
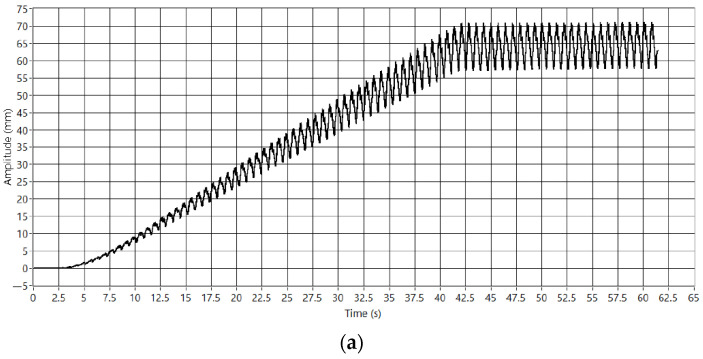
VC simulation. (**a**) CP position. (**b**) CPI force.

**Figure 11 sensors-23-01773-f011:**
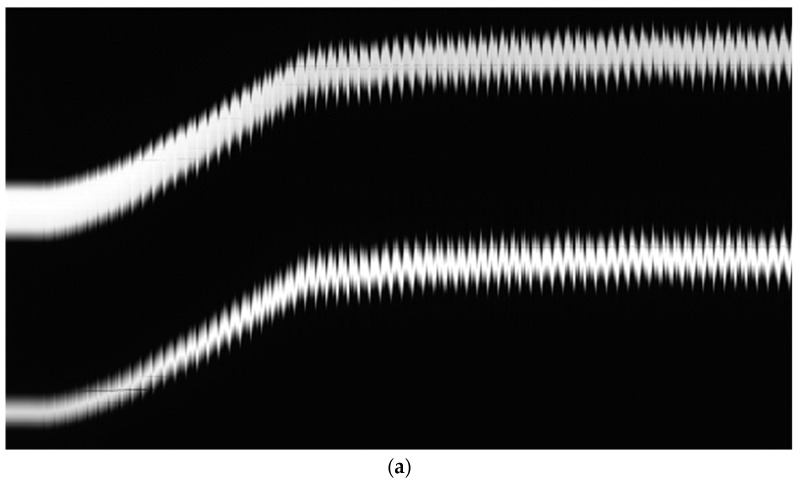
Linear camera image of HIL simulations. (**a**) HIL initialization process. (**b**) Periodic catenary behavior.

**Figure 12 sensors-23-01773-f012:**
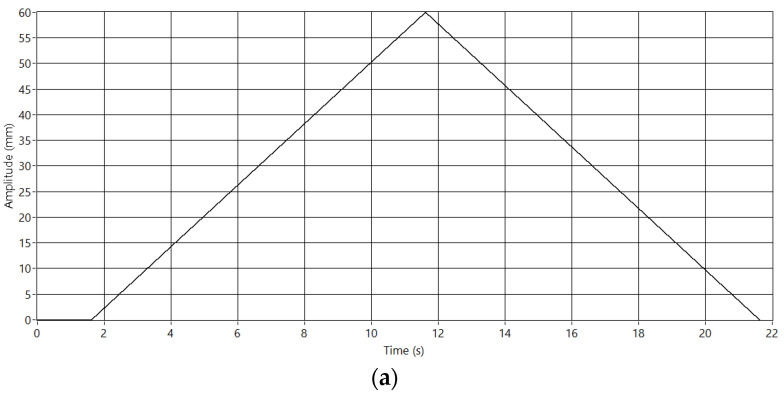
Camera calibration. (**a**) Movement pattern. (**b**) Image obtained.

**Figure 13 sensors-23-01773-f013:**
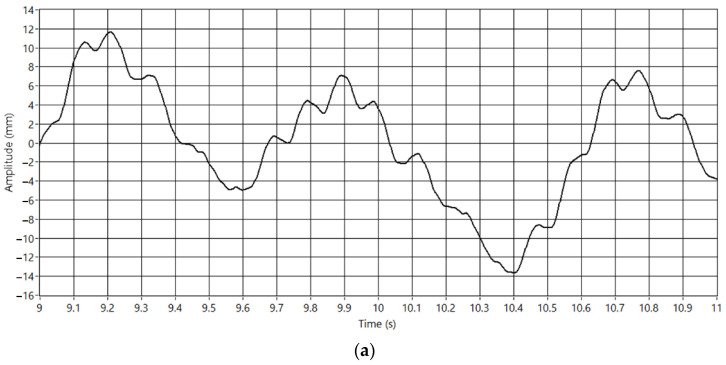
Contact loss HIL simulation. (**a**) CP position. (**b**) CPI Force.

**Figure 14 sensors-23-01773-f014:**
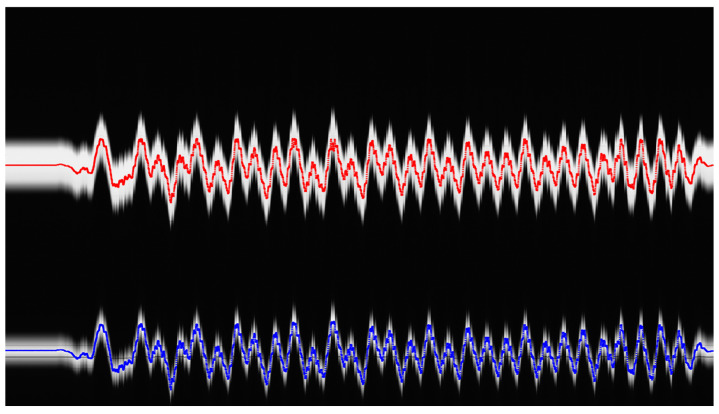
Contact loss HIL simulation centroids of the light beams.

**Figure 15 sensors-23-01773-f015:**
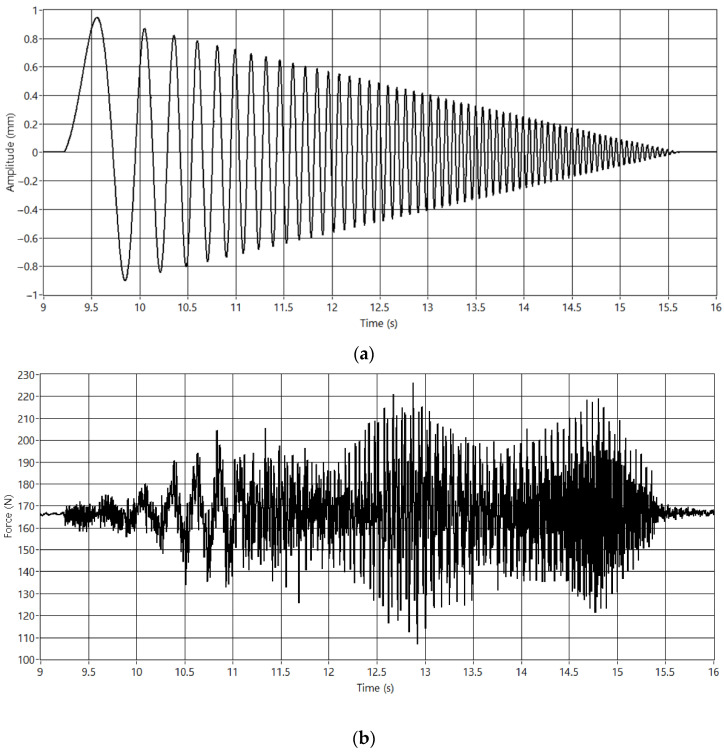
Chirp input experiment. (**a**) Movement pattern. (**b**) Force obtained.

**Table 1 sensors-23-01773-t001:** Main characteristics of the pantograph.

Parameter	Value
Max voltage	25 kV
Max current	1000 A
Compressed air supply	4–10 bar
Spring stroke	60 mm

## Data Availability

The data presented in this study are available on request from the corresponding author.

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
