# Peer review of "Hardware-in-the-Loop Test Bench for Simulation of Catenary–Pantograph Interaction (CPI) with Linear Camera Measurement"

_sensors, 2023, doi:10.3390/s23041773_

Round 1

Reviewer 1 Report

Generally, this is an innovative work to perform the HIL test of a pantograph with linear camera measurement. This paper deserves to be published with correcting the following issues.

1) In the abstract, the purpose of guaranteeing the current collection quality is not just relevant to the lifetime of the catenary. It is also important to keep a stable electric transmission to the train and avoids accidents.

2) In the introduction, another significant disturbance to PCI performance is the aerodynamic loads from the wind [1] and the high-speed flow [2]. It is recommended to add them to give the readers a complete picture of the state-of-the-art.

[1] "Wind deflection analysis of railway catenary under crosswind based on nonlinear finite element model and wind tunnel test." Mechanism and Machine Theory 168 (2022): 104608.

[2] "Pantograph aerodynamic effects on the pantograph–catenary interaction." Vehicle System Dynamics 44.sup1 (2006): 560-570.

3) It is desired to see the necessity of including computer vision technique in the HIL test of pantograph.

4) To help the reader follow this work, it is desired to directly point out what kind of quantities the linear camera measures at the beginning of Section 2. Am I right the camera measures the vertical displacements of the pantograph head and contact wire?

5) Regarding the contact loss simulation, normally, the contact loss happens at a very high frequency. Please quantify the sampling frequency in this simulation.

Reviewer 2 Report

This paper proposes an innovative measure to implement the HIL test of pantograph via a vision-based technique. This paper is well-written and well organized. It can be considered to be published with addressing the following minor issues.

Some references seem not to be very appropriate. For instance, [11-12] are not from mainstream journals, and their qualities are worrying. For the study of track irregularities’ effect on the pantograph-catenary, the following two papers have much better quality.

[1] Yao, Y., Zou, D., Zhou, N., Mei, G., Wang, J.; Zhang, W. (2021). A study on the mechanism of vehicle body vibration affecting the dynamic interaction in the pantograph–catenary system. Vehicle System Dynamics, 59(9), 1335–1354. https://doi.org/10.1080/00423114.2020.1752922

[2] Song, Y., Wang, Z., Liu, Z.; Wang, R. (2021). A spatial coupling model to study dynamic performance of pantograph-catenary with vehicle-track excitation. Mechanical Systems and Signal Processing, 151, 107336. https://doi.org/10.1016/j.ymssp.2020.107336

References [26] and [37] are repetitive. Please fix this issue.

How to achieve a real-time simulation? Very few words are found in the texts to introduce the mimic catenary. Can more explanations be compensated for?

Please clarify what exactly is captured by the camera. Is the pantograph head that can be used to calculate the vertical displacement?

Reviewer 3 Report

For title: the title can be changed to ‘Hardware in the loop test bench for simulation of Catenary-pantograph interaction (CPI) with linear camera measurement’. A common acronym can be spelt out in the title. 

Ln.26: ‘In 2019, transport consumed over 130 tons of oil equivalent.’ The sentence requires a citation. When citing the same author's work multiple times in one paragraph, you do not need to reference the author at the end of each sentence. That would look clunky and make your writing stilted. Instead, introduce the author with a full in-text citation at the beginning of the paragraph and then, again, at the end.

Ln.27 and onwards: there are another two sentences requiring citations. If they are from the same source, please see the comment above. 

Ln.31: Please use ‘often’ in a sentence. There are many forms of use of electricity for trains and trams.  

Ln.35: The meaning of sliding components varies from a jargon (British) to another one (Australian), such as the slide chair. Please give a few names of specific components in parentheses to avoid confusion.   

Ln. 120: You can end the section by providing how the rest of the paper is organized (chapters 2 , 3 and so on.). This would enhance the overall presentation of the paper. 

Ln. 154:  The real-time controller (for consistency)

Ln. 212: Please clarify the following. Each pixel takes a byte in ms. Thus, 6666.7 ?????? take 6666.7 byte in ms; magnifying  6666700 in s, which is equivalent 0.0066667 Gb per sec, as being longhand form from 1 byte = 0.000000001 Gigabyte.  It is written that ‘Therefore, the connection speed between the camera and the processing equipment must be at least 6.7 Gb/s. ‘ I think it is mb/s. 

Ln.243: footnote for the unit (parts per million)

I am of the opinion that the paper is well written and refers to what is aimed to be. I am now unable to accept your paper for publication due to the minor errors mentioned above. However, if you can address all of my comments, I would be pleased to accept a revised manuscript.
